

# Identification and analysis of lncRNA, microRNA and mRNA expression profiles and construction of ceRNA network in *Talaromyces marneffei*-infected THP-1 macrophage

Yueqi Li[1,*], Wudi Wei[1,2,*], Sanqi An[1,2], Junjun Jiang[1,2], Jinhao He[1], Hong Zhang[1], Gang Wang[2], Jing Han[1], Bingyu Liang[2], Li Ye[1,2] and Hao Liang[1,2]

[1] Guangxi Collaborative Innovation Center for Biomedicine & Guangxi-ASEAN Collaborative Innovation Center for Major Disease Prevention and Treatment, Life Science Institute, Guangxi Medical University, Nanning, Guangxi, China
[2] Guangxi Key Laboratory of AIDS Prevention and Treatment, School of Public Health, Guangxi Medical University, Nanning, Guangxi, China
* These authors contributed equally to this work.

Corresponding authors
Li Ye, yeli@gxmu.edu.cn
Hao Liang, lianghao@gxmu.edu.cn

## ABSTRACT

**Background:** Competitive endogenous RNA (ceRNA) reveals new mechanisms for interactions between RNAs, which have been considered to play a significant role in pathogen-host innate immune response. However, knowledge of ceRNA regulatory networks in *Talaromyces marneffei* (TM)-macrophages is still limited.

**Methods:** Next-generation sequencing technology (NGS) was used to obtain mRNA, miRNA and lncRNA expression profiles in TM-infected macrophages. The R package DESeq2 was used to identify differentially expressed lncRNA, miRNA and mRNA. The R package GOseq was used for Gene Ontology (GO) and Kyoto Encyclopedia of Genes and Genomes (KEGG) analysis, and the ceRNA network of lncRNA–miRNA–mRNA interaction was constructed in Cytoscape. Similarly, functional enrichment analysis on mRNA in the ceRNA network. Finally, two mRNAs and four lncRNAs in the ceRNA network were randomly selected to verify the expression using qRT-PCR.

**Results:** In total, 119 lncRNAs, 28 miRNAs and 208 mRNAs were identified as differentially expressed RNAs in TM-infected macrophages. The constructed ceRNA network contains 38 lncRNAs, 10 miRNAs and 45 mRNAs. GO and KEGG analysis of mRNA in the ceRNA network indicated that activated pathways in TM-infected macrophages were related to immunity, inflammation and metabolism. The quantitative validation of the expression of four randomly selected differentially expressed lncRNAs, AC006252.1, AC090197.1, IL6R-AS1, LINC02009 and two mRNAs, CSF1, NR4A3 showed that the expression levels were consistent with those in the RNA-sequencing.

**Conclusions:** The ceRNA network related to immunity, inflammation and metabolism plays an important role in TM-macrophage interaction. This study

may provide effective and novel insights for further understanding the underlying mechanism of TM infection.

# INTRODUCTION

*Talaromycosis* is an invasive mycosis caused by infection of a thermally dimorphic fungus *Talaromyces marneffei* (TM) that is endemic throughout Southern China, Southeast Asia, and Northeastern India (*Hu et al., 2013*; *Li et al., 2019*; *Vanittanakom et al., 2006*; *Wong, Siau & Yuen, 1999*). The number of *talaromycosis* cases has rapidly increased due to the HIV epidemic, ranking third as the most common HIV-associated opportunistic infections and accounting for up to 16% of HIV admissions and is a leading cause of death in patients with advanced HIV disease in Thailand, Vietnam, and southern China (*Hu et al., 2013*; *Jiang et al., 2019*; *Le et al., 2011*; *Sirisanthana & Supparatpinyo, 1998*). Without early diagnosis or timely antifungal treatment, The mortality in patients with disseminated disease reach 80–100%, and even with antifungal treatment, the mortality is as high as 30% (*Hu et al., 2013*; *Jiang et al., 2019*; *Le et al., 2011*; *Son, Khue & Strobel, 2014*). Therefore, a better understanding of the pathogenesis and host defense mechanisms of *talaromycosis* will help provide new and more effective diagnosis and treatment strategies.

In nature, TM grows into mycelium and produces conidia. Conidia are sucked into the host's lungs and subsequently transformed into yeast form (*Cogliati et al., 1997*). Specialized phagocytes, especially macrophages, are the first line of defense against TM infection (*Lu et al., 2013*). Traditionally, the phagocytosis and killing of TM conidia by macrophages to control the disease. Paradoxically, previous results reported that TM conidia can germinate and survive in macrophages (*Lu et al., 2013*; *Vanittanakom et al., 2006*). From the perspective of the pathogen, TM has been showed to evolve strategies to cope with the harsh intracellular environment of macrophages, such as overcoming the dilemma of nutritional deprivation through glyoxylate cycle, converting to yeast form to adapt to heat stress. However, little is known about the changes in macrophage gene expression profile after TM infection. The study of changes in the expression of these genes may help to further understand the regulation of anti-TM immunity in macrophages. More importantly, new biomarkers for *talaromycosis* diagnostic and therapeutic targets may also be discovered.

mRNA and lncRNA are important components in gene expression profile changes caused by pathogen–macrophage interactions. The function of mRNA can be revealed through Gene Ontology (GO) and Kyoto Encyclopedia of Genes and Genomes (KEGG) analysis, but the function of lncRNA is largely unknown. In the past few years, the research interest of lncRNA has increased dramatically, which has attracted wide attention as a potential and important biological regulatory factor. LncRNAs are defined as the transcripts with a length longer than 200 nt and without protein-coding capacity.

It is generally believed that lncRNAs regulate gene expression by acting as transcription coactivators, RNA decoys, or microRNA sponges (*Hung & Chang, 2010*). Some studies have reported that lncRNA plays a vital role in many biological processes (*Aune & Spurlock, 2016*; *Moran, Perera & Khalil, 2012*; *Prensner & Chinnaiyan, 2011*; *Satpathy & Chang, 2015*), and recent studies have also suggested that lncRNAs have the potential to serve as new biomarkers or therapeutic targets for some diseases (*Boon et al., 2016*). Several studies have reported that some lncRNAs, such as MEG3 (*Pawar et al., 2016*), lincRNA-EPS (*Atianand et al., 2016*) and Lnczc3h7a (*Lin et al., 2019*) play important roles in autophagy, inflammation, and immune responses in macrophages. However, the information about the role of lncRNA-related Competitive endogenous RNA (ceRNA) network in the TM-macrophage interactions is limited.

In this study, a TM-infected macrophage model was constructed and the expression profiles of lncRNA, mRNA and miRNA in TM-infected macrophages were exhibited by high-throughput sequencing. Compared with uninfected macrophages, the expression profiles of lncRNA and mRNA in TM-infected macrophages changed significantly. The ceRNA network was constructed based on differentially expressed RNA. GO and KEGG analyses of mRNA in ceRNA network showed that TM infection of macrophage activate immune, metabolic and inflammatory responses. Overall, the present study indicates that TM infection of macrophages significantly affect the expression of lncRNA and mRNA, which may regulate the functions of immune and inflammatory responses of TM-infected macrophages through the ceRNA network.

## MATERIALS AND METHODS

### Cell line, fungal strain and TM infection

The human monocyte cell line THP-1 was purchased from the cell bank of the Chinese Academy of Sciences, and TM strain American Type Culture Collection (ATCC) 18224 was purchased from ATCC.

RPMI 1640 medium containing 10% FBS, 100 U/mL penicillin and 100 µg/mL streptomycin was used for cultures of THP-1 cells at 37 °C with 5% $CO_2$. Medium was refreshed every three days. After 72 h of stimulation with 50 ng/ml PMA, THP-1 cells were differentiated into macrophages, which were maintained in 10% heat-inactivated FBS in DMEM containing 100 U/mL penicillin and 100 µg/mL streptomycin for further processing.

The TM strain ATCC18224 was inoculated on potato dextrose agar (PDA) agar, conidia were obtained after 7–10 days' incubation at 27 °C. The conidia were separated and purified by filtration through sterile glass wool. The conidia suspension at a concentration of $10^7$ conidia/ml was prepared for subsequent experiments.

For the TM infection, THP-1 macrophages were co-cultured with TM purified conidia (MOI = 10) for 24 h.

In this study, at least three replicates were set for both the TM-infection and control groups.

## Total RNA extraction

Total RNA from the samples was isolated using Trizol reagent according to the manufacturer's instructions (Invitrogen, Carlsbad, CA, USA). RNA integrity was detected using an RNA Nano 6000 Assay Kit from the Agilent Bioanalyzer 2100 system (Agilent Technologies, Santa Clara, CA, USA). RNA purity and concentration was measured using the NanoPhotometer® spectrophotometer (IMPLEN, CA, USA). The 260/280 ratio for all samples was approximately 2.0 and the RNA integrity number (RIN) ≥ 8.0.

## Differentially expressed lncRNAs and mRNAs in macrophages

Clean data were obtained by trimming reads containing adapter, reads containing over 5% of ploy-N and low-quality reads (> 50% of bases whose Phred scores were <10) from the raw data by using Perl scripts. Human reference genome (Ensembl GRCh38.95) and gene annotation file are downloaded from Ensembl (https://asia.ensembl.org/index.html). Clean reads were aligned to the reference genome using HISAT2 (v2.1.0) (https://daehwankimlab.github.io/hisat2/). The mapped reads were assembled using Stringtie (v 1.3.3) (https://ccb.jhu.edu/software/stringtie/) in a reference-based approach. Based on the transcripts reconstructed by Stringtie, the transcripts reconstructed from multiple samples are first compared with each other through the cuffcompare software, and a unique set of transcripts is obtained by removing redundancy. Finally, a series of filters were performed on the reconstructed transcript to obtain sequencing information of lncRNA and mRNA, and the encoding ability of the filtered transcript was predicted by lncRNA prediction software (CPC, CNCI and Pfam). The crossover results were combined with known lncRNAs for subsequent analysis. We use DESeq2 (https://bioconductor.org/packages/release/bioc/html/DESeq2.html) for differential expression analysis, and set the threshold for the significant differential expression of lncRNA and mRNA to $P$-value < 0.05 and |Fold of change| ≥ 1.5.

## Small RNA extraction

We first extracted the total RNA of the sample, and then separated and recovered small RNA with a size of 18–30 nt using PAGE gel. Then, the recovered small RNA was mixed and centrifuged in a 3'ligation system. After keeping it at a suitable temperature for a certain period of time, add a 5′connection system. Then the 5′linker was reverse transcribed into double strands by adding a reverse transcription system, and finally purified by qRT-PCR amplification and PAGE gel recovery.

## Differentially expressed miRNA in macrophages

The raw reads obtained by the sequencing platform are passed through data quality control (QC) and filtered to obtain high quality clean tags. The software Bowtie2 (http://bowtie-bio.sourceforge.net/bowtie2/index.shtml) was used to map clean tags to the human genome (Ensembl GRCh38.95). Differential analysis was performed using

DESeq2, and differential miRNA screening conditions were: |Fold of change| ≥ 1.5 and $P$-value ≤ 0.05.

## GO and KEGG analysis

GO term enrichment analysis to obtain gene annotations and functional enrichment information, including BP (biological process), CC (cell component) and MF (molecular function). Then the R package Goseq is used to perform GO enrichment analysis. The software calculates the $P$-value through the Wallenius non-central hyper-geometric distribution. We set $P$-value < 0.05 as the significance threshold to obtain statistically significant high-frequency annotations relative to the control group.

The KEGG pathway annotation is used to classify differentially expressed genes by biological pathways. $P$-values were calculated using Fisher's exact test. $P$-value < 0.05 was used as a threshold to determine the statistical significance of signal transduction and disease pathway enrichment. KEGG pathway enrichment of differentially expressed mRNAs was performed by the Goseq package on the R platform .

## Gene Set Enrichment Analysis (GSEA)

Gene Set Enrichment Analysis (GSEA) is a computational method that determines whether an a priori defined set of genes shows statistically significant, concordant differences between two biological states. The sequencing data was enriched using GSEA-3.0.jar (https://software.broadinstitute.org/gsea/index.jsp). The number of random combinations was 1,000. For the analysis results, it is considered that the gene set under the pathway of |NES| > 1, NOM $P$-value < 0.05, and FDR $q$-value < 0.25 is significant.

## ceRNA network analysis

We used the miRNA target gene prediction software miRanda (v3.3a) (www.microrna.org), TargetScan (v7.2) (http://www.targetscan.org/vert_72/) and miRTarBase (v8.0) (http://mirtarbase.cuhk.edu.cn/php/index.php) to predict miRNA target genes and obtain the intersection of the three databases. Since mRNA has a wide range of regulatory effects, we extend targeted mRNAs to the entire database instead of our sequencing data. Then, based on the prediction results of lncRNA–miRNA and miRNA–mRNA targeting pairs, a ceRNA network was organized and generated, which was finally visualized by Cytoscape (v3.7.1).

## qRT-PCR

Total cellular RNA was isolated using Trizol reagent, and cDNA was synthesized by reverse transcription using the DRR036A Takara PCR kit according to the manufacturer's instructions. Subsequently, we performed qRT-PCR using SYBR Green as previously described. Human housekeeping GAPDH was used as a reference. For quantitative analysis, a $2^{-\Delta\Delta Ct}$ method was used to calculate the relative expression level of each lncRNA and mRNA. The data are expressed as the mean ± SD, with $P$-value < 0.05 as the considered significance (Student's $t$-test). Table 1 lists PCR primers used in this study.

**Table 1  Primers for qRT-PCR.**

| mRNA | Forward (5′–3′) | Reverse (5′–3′) |
|---|---|---|
| CSF1 | TGGCGAGCAGGAGTATCAC | AGGTCTCCATCTGACTGTCAAT |
| NR4A3 | TGCGTCCAAGCCCAATATAGC | GGTGTATTCCGAGCTGTATGTCT |
| **lncRNA** | **Forward (5′–3′)** | **Reverse (5′–3′)** |
| IL6R-AS1 | GCATGGACGGACAGAGCTTC | GCTCACAGTCCCTCTCTGGT |
| LINC02009 | AGAGAGATAAATGCAGCGTGGTC | TGCTCTTTGCAAGACAGTGCC |
| AC090197.1 | CTCCACGTAGCCCTCCATCA | CTGTCGTTTGAGCCCTGACC |
| AC006252.1 | TACCTGCCTGCTGCTACCAA | TAGGCCCAGTCTTCAGGGTG |

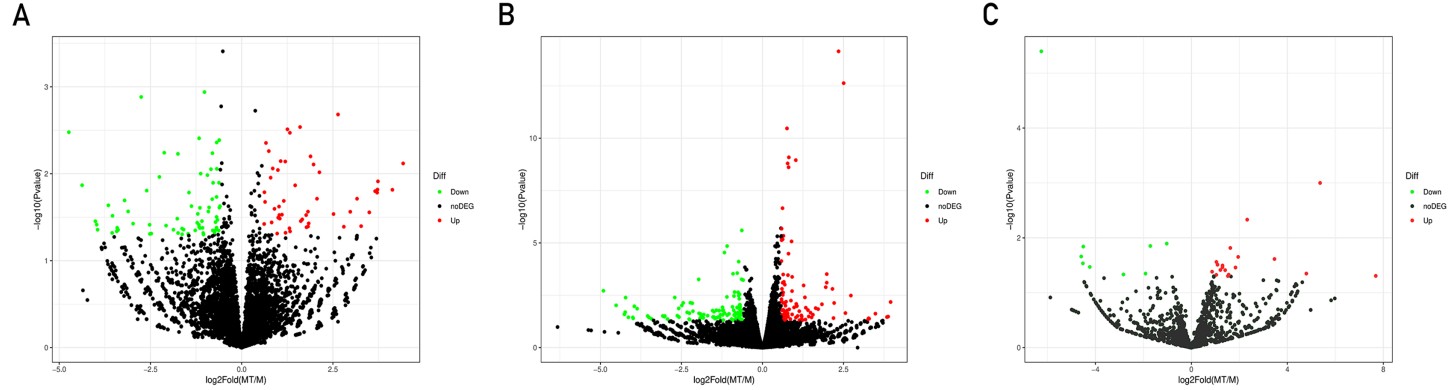

**Figure 1  Volcano plots of differentially expressed lncRNAs (A), mRNAs (B) and miRNAs (C).** Each point represents a gene. The green dots denote down-regulated DEGs, the red dots denote up-regulated DEGs, and the black dots denote non-DEGs. The *X*-axis represents the log (base 2) of fold of change (MT/M), and the *Y*-axis represents the negative log (base 10) of the *P*-value. DEGs, differentially expressed genes. MT, TM-infected macrophages. M, control group.                                               

## RESULTS

### Differentially expressed lncRNA, mRNA and miRNA profiles

In order to explore the molecular profile of human macrophage after TM infection, we examine the quantity and sequences of RNA in macrophages using Next-generation sequencing (NGS).

A total of 21,585 mRNAs, 7,881 lncRNAs and 1,677 miRNAs were obtained. By setting the filter condition of |Fold of change| ≥ 1.5 and *P*-value < 0.05, a total of 208 differentially expressed mRNAs (109 upregulated and 99 downregulated), 120 differentially expressed lncRNAs (50 upregulated and 70 downregulated) and 29 differentially expressed miRNAs (20 upregulated and nine downregulated) were obtained. Volcano map of differential gene expression is shown in Fig. 1.

The clustering heatmaps are shown in Fig. 2. Hierarchical clustering shows that lncRNAs, miRNAs and mRNAs expression patterns among samples were distinguishable.
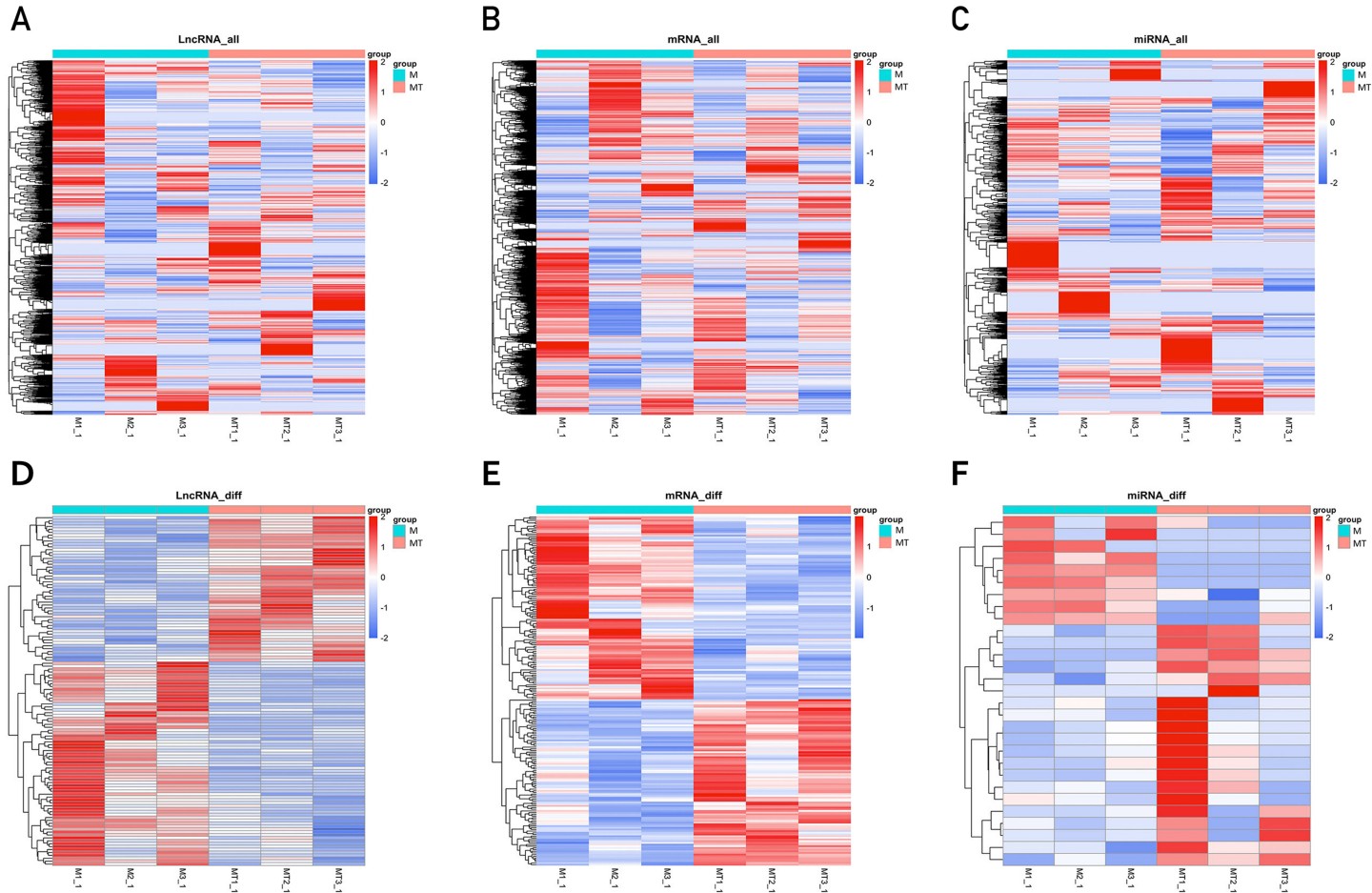

**Figure 2 Clustering heat maps of differentially expressed lncRNA (A and D), mRNA (B and E) and miRNA (C and F).** A–C are based on expression values of all lncRNAs, mRNAs, and miRNAs that were detected, while D–F showed the expression values of significantly changed lncRNAs, mRNAs and miRNAs. The expression values are depicted in line with the color scale. The intensity increases from blue to red. Each column represents one sample, and each row indicates a gene. The values in the heat map is the *z*-score of log2 (FPKM+1).

## Delineation of GO, KEGG pathway and GSEA analysis of mRNA

The Goseq package in the R platform was used for GO and KEGG analysis of differentially expressed mRNA. Calculate and return the *P*-value by GO and KEGG analysis based on the differential gene. The statistic significant threshold value for enrichment analysis was *P*-value < 0.05. In the BP category, we selected 10 most significantly enriched terms for display, which are related to immunity and metabolism. The CC and MF category show the most significant enriched terms in molecular-level activities and locations of differentially expressed RNA. In this study, we focused more on GO terms related to macrophage immune response, inflammatory response, and metabolic processes. Figure 3A shows that differentially expressed mRNA is significantly enriched in some BP associated with immune and inflammatory responses which are overlapped with the pathways we are interested in. Functional enrichment results are consistent with expectations.

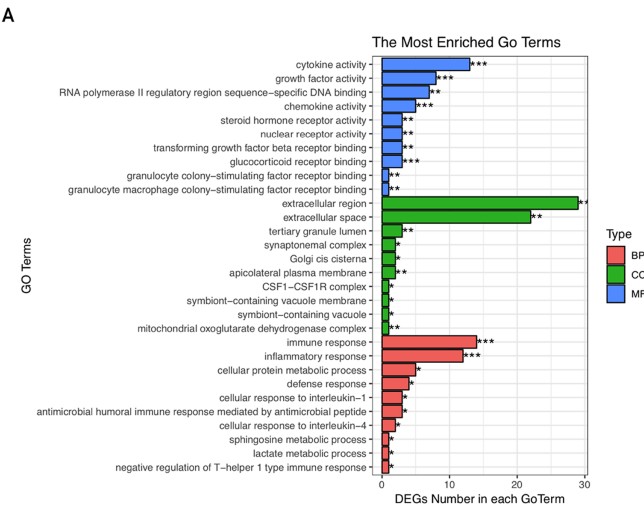

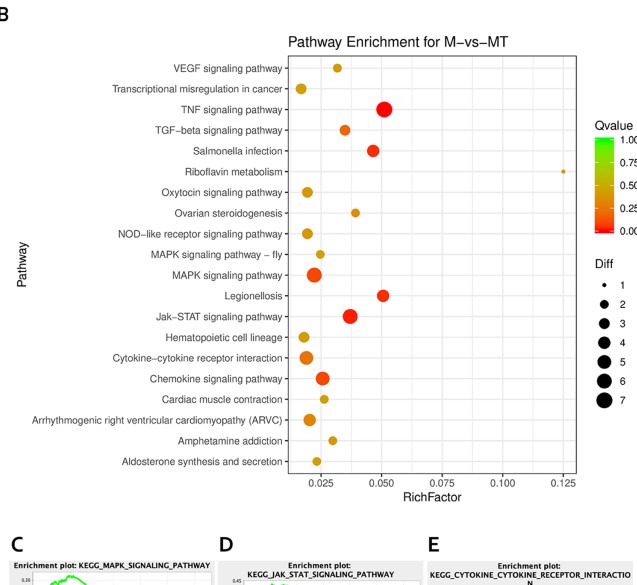

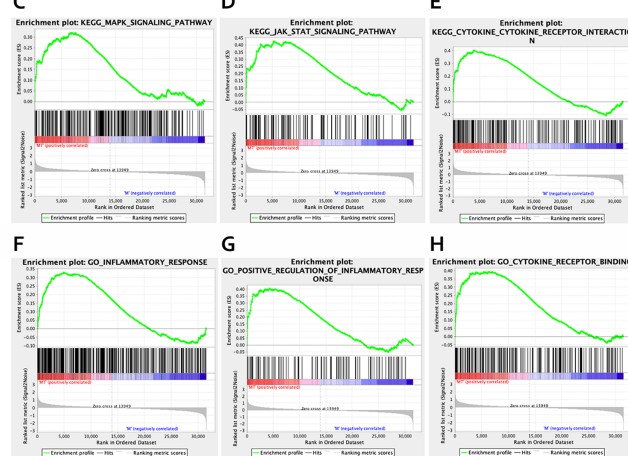

**Figure 3 GO and KEGG analyses of differentially expressed genes.** (A) The *Y*-axis represents the enriched GO term, and the *X*-axis represents the number of genes that are significantly different in the GO term. Different colors are used to distinguish biological processes, cellular components, and molecular functions. *, $P < 0.05$; **, $P < 0.01$; ***, $P < 0.001$. (B) The *Y*-axis indicates the pathway entry, the *X*-axis indicates the rich factor; the dots indicates the significant differential expression of the gene
**Figure 3** (continued)
number, the size of the dot indicates the number of significant differentially expressed genes, and the dots
of different colors indicate different *q*-value. (C–H) The name of the pathway is shown above the figures,
and the ES peak appears at the front to indicate that the pathway is activated in the group and vice
versa.                               

The KEGG pathway enrichment analysis was used to explore which significant pathways were activated in TM-infected macrophage. The top 20 enriched pathways included TNF signaling pathway, Riboflavin metabolism, NOD–like receptor signaling pathway, MAPK signaling pathway, JAK–STAT signaling pathway, Cytokine–cytokine receptor interaction, and so on. The enriched KEGG pathways is shown in Fig. 3B.

From the perspective of gene set enrichment, GSEA is theoretically easier to include the effects of subtle but coordinated changes on biological pathways. Therefore, we use all mRNA expression matrices for GSEA enrichment analysis. As shown in Fig. 3C, compared to control (group M), the MAPK signal pathway, the JAK-STAT signal pathway, and the Cytokine - Cytokine receptor pathway of the TM-infected macrophages (group MT) are all activated.

## ceRNA network

We use miRNA target gene prediction software miRanda (v3.3a), TargetScan (v7.2) and miRTarBase (v8.0) to predict miRNA-lncRNA and miRNA-mRNA respectively, and then intersect the prediction results of the three databases. After strict filtration, the mRNA and lncRNA with the same miRNA target are counted. Finally, there are 38 unregulated lncRNAs (18 up-regulated and 20 down-regulated lncRNAs), 45 mRNAs (28 up-regulated, three down-regulated and 14 undifferentiated mRNAs) and 10 unregulated miRNAs (six up-regulated and four down-regulated miRNAs) are integrated into the ceRNA network and 679 connections are established, as shown in Fig. 4. Table 2 lists the RNAs in ceRNA.

In order to explore the role of lncRNA in the ceRNA network, we conducted GO and KEGG analysis on the target mRNA of lncRNA, and then explored the function of lncRNA.

The results of GO and KEGG analysis are shown in Figs. 5A and 5B. GO enrichment analysis results showed that these BP pathways that are competitively regulated by lncRNA are significantly enriched in neutrophil chemotaxis, regulation of signaling, receptor activity cytokine–mediated signaling pathway, inflammatory response, cellular response to lipopolysaccharide, corticotropin–releasing hormone. In terms of stimulus, immune response, etc. KEGG results showed that TNF signaling pathway, MAPK signaling pathway, JAK-STAT signaling pathway, Chemokine signaling pathway and other pathways are the main functional pathways for lncRNA to regulate mRNA competitively.

## qRT-PCR validation

In order to evaluate the accuracy of the analysis results, we randomly selected four lncRNAs, AC006252.1, AC090197.1, IL6R-AS1, LINC02009 and two mRNAs, CSF1,

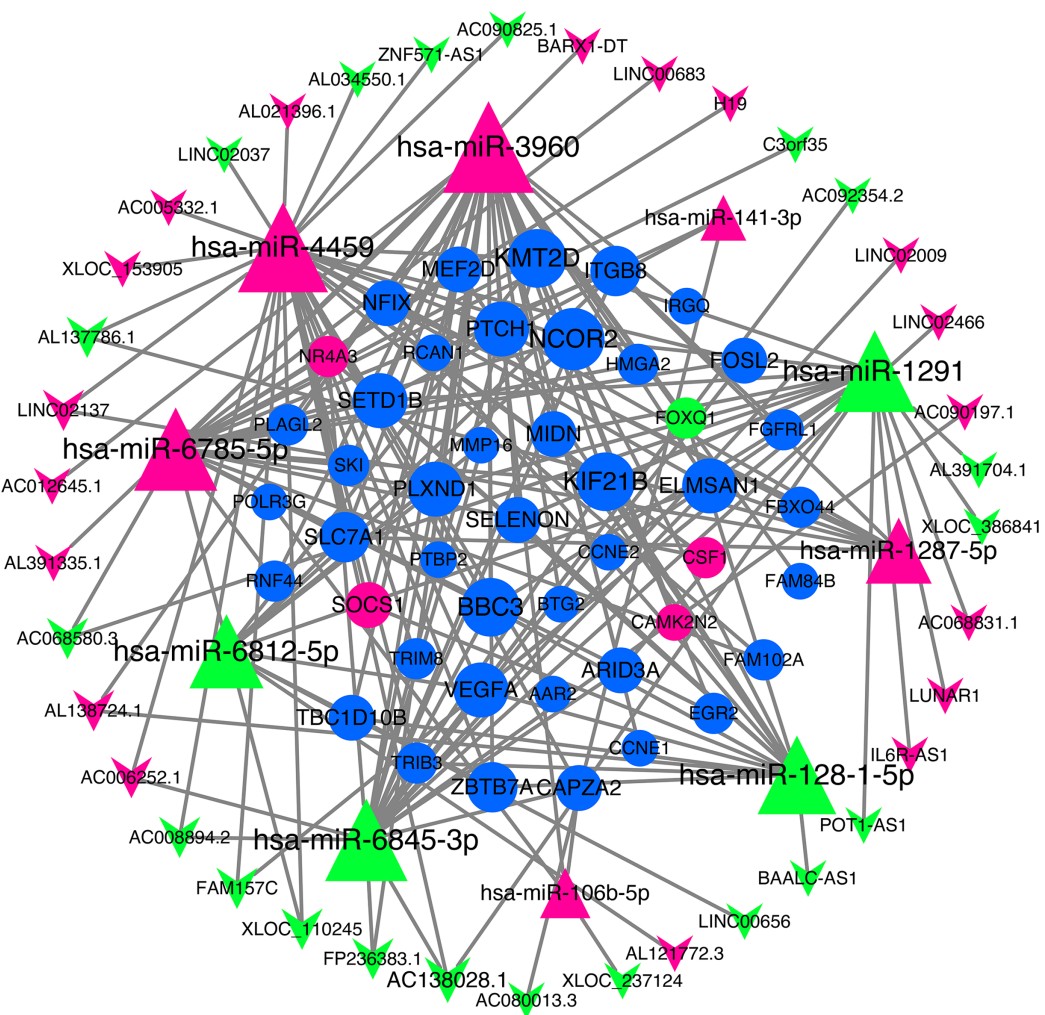

**Figure 4 lncRNA-miRNA-mRNA networks.** The triangle represents miRNA, V-shape figure represents lncRNA, circle represents mRNA. down-regulated expression gene were marked green, up-regulated expression gene were marked red, the genes that have a targeted relationship in public databases although there is no significant difference in this experiment were marked blue. Large nodes indicate high connectivity and vice versa.

**Table 2 The nodes in lncRNA-miRNA-mRNA networks.**

| Type | Symbol |
|------|--------|
| LncRNA | H19, AC138028.1, C3orf35, AL391704.1, POT1-AS1, IL6R-AS1, AL021396.1, LINC00656, AC068580.3, BARX1-DT, AL034550.1, LINC02037, AC006252.1, LINC02466, BAALC-AS1, AC012645.1, AL391335.1, AC090197.1, AC068831.1, AL137786.1, AC090825.1, LINC02137, FAM157C, AC005332.1, LINC00683, ZNF571-AS1, AC008894.2, AC080013.3, AL138724.1, AC092354.2, AL121772.3, LUNAR1, FP236383.1, LINC02009, XLOC_110245, XLOC_153905, XLOC_237124, XLOC_386841 |
| mRNA | SYMBOL, PTCH1, POLR3G, CAPZA2, VEGFA, NR4A3, MEF2D, TRIB3, SELENON, NCOR2, FOXQ1, TBC1D10B, BBC3, SOCS1, NFIX, KIF21B, ZBTB7A, SETD1B, MIDN, ITGB8, SLC7A1, ELMSAN1, FGFRL1, CCNE2, KMT2D, ARID3A, PLXND1, RCAN1, BTG2, LRATD2, PLAGL2, PTBP2 |
| miRNA | miR-106b-5p, miR-128-1-5p, miR-1287-5p, miR-1291, miR-141-3p, miR-3960, miR-4459, miR-6785-5p, miR-6812-5p, miR-6845-3p |

A
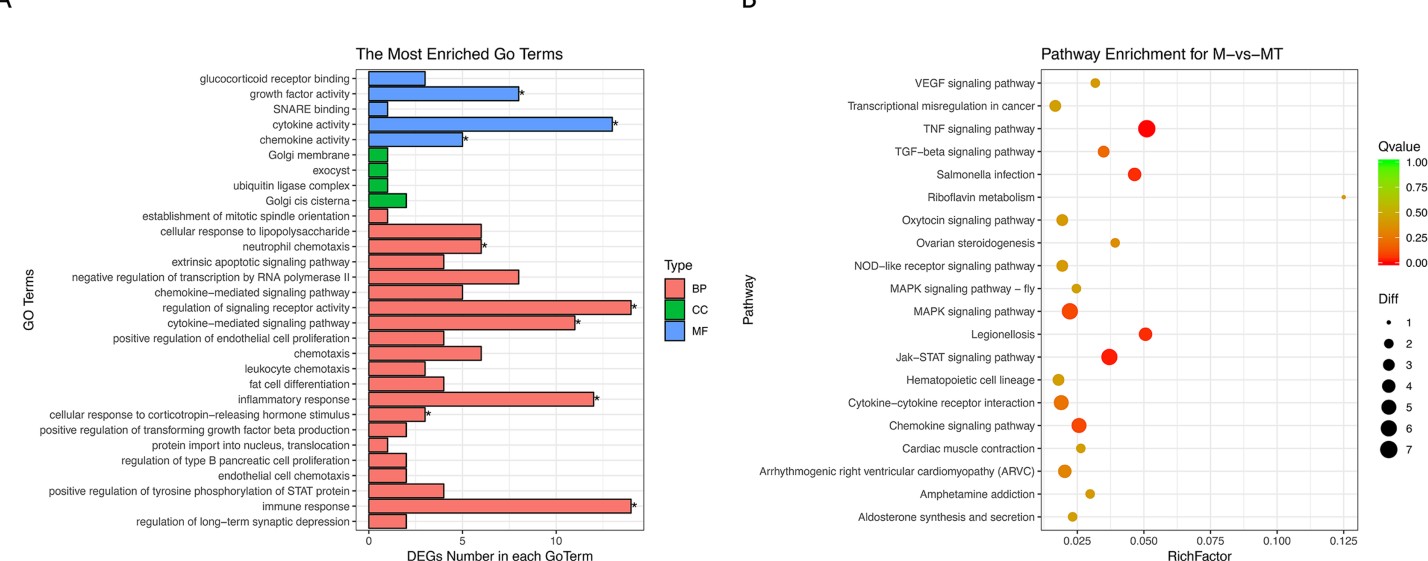
B

**Figure 5 GO and KEGG analyses of differentially expressed genes in ceRNA network.** (A) The *Y*-axis represents the enriched GO term, and the *X*-axis represents the number of genes that are significantly different in the GO term. Different colors are used to distinguish biological processes, cellular components, and molecular functions. *, $P < 0.05$. (B) The *Y*-axis indicates the pathway entry, the *X*-axis indicates the rich factor; the dots indicate the significant differential expression of the gene number, the size of the dot indicates the number of significant differentially expressed genes, and the dots of different colors indicate different *q*-value values.

NR4A3 for qRT-PCR verification. The results of qRT-PCR were consistent with those of RNA-Seq (Fig. 6).

## DISCUSSION

In this study, we investigated the expression patterns of lncRNA and mRNA in human THP-1 macrophages infected with TM. Compared with the uninfected macrophages, the lncRNAs and mRNAs expression profiles of TM-infected macrophage changed significantly, most were related to immune and inflammatory responses.

Totally, 208 mRNAs, 120 lncRNAs and 29 miRNAs were differentially expressed between TM-infected and control macrophages. Compared to the control, 109 mRNA expressions were up-regulated and 99 expressions were down-regulated in the TM-infected macrophage. The GO and KEGG analyses showed that the differentially expressed mRNA are mainly related to immune and inflammatory response pathways, including the TNF signaling pathway, Riboflavin metabolism, MAPK signaling pathway, JAK-STAT signaling pathway, Cells factor-cytokine receptor interactions, and so on. GSEA analysis proved that the MAPK signaling pathway is activated by TM infection, which result in numerous physiological activities, such as inflammation, apoptosis, canceration, tumor cell invasion and metastasis. In macrophages, the MAPK signaling pathway is a key signaling pathway that regulates the host immune response to infection. Activation of the MAPK signaling pathway may enhance the immune phagocytosis and inflammatory response of macrophages. p38 subfamily is crucial for the initiation of the inflammatory responses (*Dong, Davis & Flavell, 2002*; *Ono & Han, 2000*).

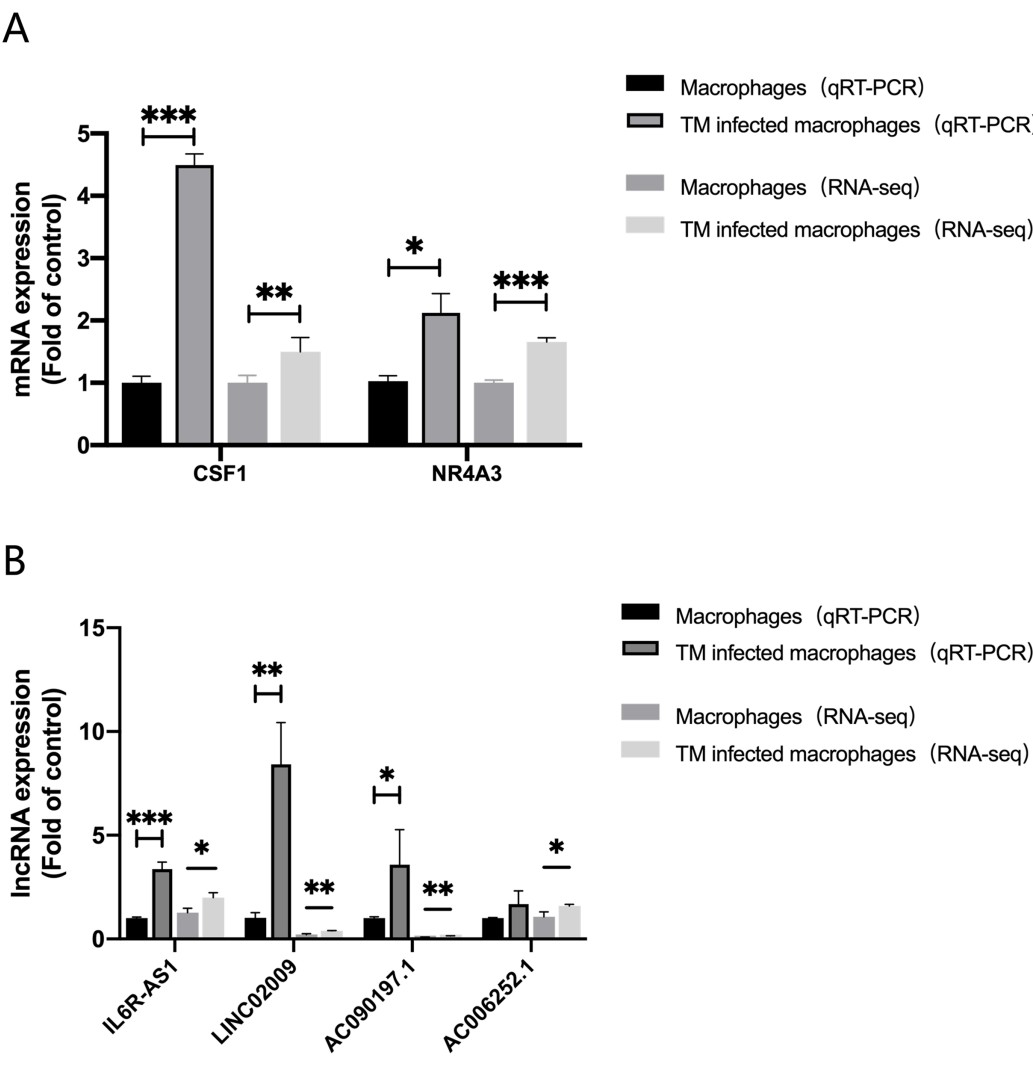

**Figure 6 qRT-PCR and RNA-seq results.** (A) Expression of mRNA CSF1 and NR4A3. (B) Expression of lncRNA IL6R-AS1, LINC02009, AC090197.1 and AC006252.1. *, $P < 0.05$; **, $P < 0.01$; ***, $P < 0.001$.

Our sequencing data indicated that the expression of MKP, HSP72, GADD45 and Nur77 in the MAPK signaling pathway were all increased, while the expression of CACN were down-regulated. These factors are important in the inflammatory response, however, current understanding of the key role of these factors in macrophages is still limited. A research indicates that overexpression of Nur77 inhibits macrophage apoptosis and increases cell survival by inhibiting apoptosis without affecting the cell proliferation of activated macrophages (*Shao et al., 2016*). The JAK-STAT signal pathway is a cytokine-stimulated signal transduction pathway discovered in recent years and is involved in many important biological processes such as cell proliferation, differentiation, apoptosis, and immune regulation. The JAK-STAT pathway related factors, such as SOCS1 (FC = 1.61, *P*-value = 0.02), CSF3 (FC = 1.71, *P*-value = 0.02), IL24 (FC = 2.06, *P*-value = 0.01) and LIF (FC = 1.71, *P*-value = 0.00) is up-regulated and

play a key role in regulating M1 and M2 polarization in human THP-1 macrophages (*McCormick & Heller, 2015*). It has been reported that CSF3 has anti-inflammatory and promotes M2 polarization in mice and human macrophages (*Hollmen et al., 2016*; *Hori et al., 2019*). TNF signaling pathway have an effect on many aspects of macrophage function, and plays an important role in macrophage inflammation, proliferation, apoptosis and polarization. The ability of macrophages to fight against TM infection is related to their polarization status. M1 macrophages have stronger phagocytosis and killing ability than M2 macrophages. M1 macrophages highly express TNF-α and low IL-10, while M2 macrophages have the opposite. The results showed that TNF-α was low-expressed (FC = 0.21, *P*-value = 0.55) and IL-10 was highly expressed (FC = 3.38, *P*-value = 0.07) in TM-infected macrophages, indicating that TM infection may induce the THP-1 macrophage M2 polarization, which may be related to the mechanism of TM avoiding macrophage killing.

lncRNAs have been shown to be involved in a variety of biological regulatory functions, including immune response, inflammatory response, polarization, and apoptosis in macrophages (*Atianand et al., 2016*; *Aune & Spurlock, 2016*; *Boon et al., 2016*; *Lin et al., 2019*; *Pawar et al., 2016*; *Satpathy & Chang, 2015*; *Zhang et al., 2020*). In this study, a total of 120 lncRNAs (50 upregulated and 70 downregulated) were identified, of which 38 lncRNAs (18 up-regulated and 20 down-regulated lncRNAs) were constructed the ceRNA network. Although the current research on lncRNAs and ceRNA networks is a hot topic, reports on its role in TM infection are still limited. Previous research have indicated that lncRNA IL6R-AS1, AC006252.1, LINC02466 were involved in immune pathways in a variety of cancers. However, the specific immune mechanism has not been reported yet, so further research is needed. Therefore, fully exploring the role of these lncRNAs induced by TM infection may be provide new insights into the pathogenesis of TM infection. In addition, with the help of functional analysis of mRNAs in ceRNA, we can understand the potential role of lncRNAs in the ceRNA network. As mentioned above, in the ceRNA networks composed of lncRNA–miRNA–mRNA, the functional analysis of ceRNA mRNA showed similar results to the above differential mRNAs, which showed that TM-related pathways are actually not only affected by differentially expressed lncRNAs, but also regulated by downstream mRNA.

In summary, our study indicates that during TM infection of macrophages, the expression profiles of lncRNAs and mRNAs changed significantly. Furthermore, lncRNAs may play an important role in TM-macrophage interactions. However, although the significant changed lncRNA and mRNA profiles due to TM infection were observed as well as a ceRNA network was established, the underlying molecular mechanism of TM-macrophage interactions still needs further study.

## CONCLUSIONS

The ceRNA network plays an important role in the mechanism of TM infection in macrophages. This study may provide effective and novel insights for further understanding of underlying mechanism of *Talaromyces marneffei*.

### Funding

The study was supported by the National Natural Science Foundation of China (NSFC; 81971934, 81760602, 31970167), the Guangxi Bagui Scholar (to Junjun Jiang), the Thousands of Young and Middle-aged Key Teachers Training Program in Guangxi Colleges and Universities (to Junjun Jiang), and the Guangxi Medical University Training Program for Distinguished Young Scholars (to Junjun Jiang). The funders had no role in study design, data collection and analysis, decision to publish, or preparation of the manuscript.

### Grant Disclosures

The following grant information was disclosed by the authors:
National Natural Science Foundation of China (NSFC): 81971934, 81760602 and 31970167.
Guangxi Bagui Scholar.
Guangxi Medical University Training Program.

### Competing Interests

The authors declare that they have no competing interests.

### Author Contributions

- Yueqi Li analyzed the data, prepared figures and/or tables, authored or reviewed drafts of the paper, and approved the final draft.
- Wudi Wei analyzed the data, prepared figures and/or tables, authored or reviewed drafts of the paper, and approved the final draft.
- Sanqi An performed the experiments, analyzed the data, authored or reviewed drafts of the paper, and approved the final draft.
- Junjun Jiang conceived and designed the experiments, authored or reviewed drafts of the paper, and approved the final draft.
- Jinhao He performed the experiments, prepared figures and/or tables, authored or reviewed drafts of the paper, and approved the final draft.
- Hong Zhang performed the experiments, prepared figures and/or tables, authored or reviewed drafts of the paper, and approved the final draft.
- Gang Wang performed the experiments, authored or reviewed drafts of the paper, and approved the final draft.
- Jing Han performed the experiments, authored or reviewed drafts of the paper, and approved the final draft.
- Bingyu Liang performed the experiments, authored or reviewed drafts of the paper, and approved the final draft.
- Li Ye conceived and designed the experiments, authored or reviewed drafts of the paper, and approved the final draft.
- Hao Liang conceived and designed the experiments, authored or reviewed drafts of the paper, and approved the final draft.
## Data Availability

Data is available at NCBI GEO: GSE154780.

## Supplemental Information

Supplemental information for this article can be found online at http://dx.doi.org/10.7717/peerj.10529#supplemental-information.

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
