# Peer review of "Identification and analysis of lncRNA, microRNA and mRNA expression profiles and construction of ceRNA network in Talaromyces marneffei-infected THP-1 macrophage"

_PeerJ, doi:10.7717/peerj.10529_

## Round 0.1 · original submission · Minor Revisions

Dear authors,

Thank you for submitting your work to PeerJ. Your work was evaluated by three independent reviewers. In order for your manuscript to be further considered for publication, I would like you to incorporate and address the comments raised by the reviewers. Once you have done that, please re-submit your work. Please pay special attention to the use of proper English grammar and style and make sure all your figures are of high quality.

Also, make sure you make publicly available the primary data used in this work by submitting the sequencing data to the GEO/SRA or equivalent databases. The same recommendation goes for any scripts/code used in your analysis. I am looking forward to your revised manuscript.

Reviewer 1 ·

Basic reporting

Li and colleagues presented a work to study the ceRNA network in macrophages infected with Talaromyces marneffei (TM). By sequencing the TM-infected macrophages and normal macrophages, they quantified the expression levels of lncRNA, miRNA and mRNA, identified the differentially expressed genes and enriched function categories, and constructed the lncRNA–miRNA–mRNA interaction ceRNA networks. Their work revealed some lncRNA markers that may play important regulatory role in TM-macrophage interactions. Please see below for some major and minor comments.

Major comments:

1. There are quite a few grammar issues with the manuscript. Here I just listed few examples. The author should carefully check the manuscript if it is accepted for publish.

(1) in abstract line 55-56, “Select genes that are significantly abundant in GO analysis and KEGG pathway. The core genes…” Technically this is not a complete sentence. The “core genes” in the next sentence is also confusing in this context.

(2) line 80-82: “If not diagnosed early or appropriate systemic antifungal treatment is not performed in time,…”, maybe “Without early diagnosis or timely systemic antifungal treatment… ” would be better?

(3) Line 167-169: technically, these two sentences should not be connect with comma. Should be rephrased.

(4) Line 172-176: all the methods used the past tense other than this part. It would be better to keep the context consistent.

(5) Line 186-188: incomplete sentence without subject.

2. Line 89-91: “It has been found that the function of macrophages on TM, including phagocytosis, fungicidal activity, and phagosome-lysosomal fusion, won’t be affected with the weakened immunity.” Is there any reference paper to support this finding?

3. Line 93-94: “The interaction between macrophages and TM changes the expression profile of cellular genes.” Is there any reference for this?

4. Line 91-94: in this part, the author mentioned the paradoxical phenomenon that TM conidia can survive in macrophages. Have there any other studies been looked into this? Did they provide any insights and if so, how would that related with this study? I would suggest the author to elucidate more details and background here, in order to state a stronger rational and motivation of this study.

5. Line 241-245: “The upper panels are based on expression values of all lncRNAs, mRNAs, and miRNAs detected. While, the bottom panels correspond to expression values of significantly changed lncRNAs, mRNAs, and miRNAs. The expression values are depicted in line with the color scale. The intensity increases from blue to red. Each column represents one sample, and each row indicates a gene.”

I would suggest move this description to the caption of Figure 2 since it’s more about how to read a plot. Also, in Figure 2, it might be better to label group a and b with specific names. The values in the heatmap should be clarified as well. I assumed it should be the z-scores of log2(FPKM)?

6. Line 272-275: “The highest peak in the Figure 3(C) is the enrichment score (ES). A positive value ES means that the gene set is enriched at the top of the list, and a negative value ES means that the gene set is enriched at the bottom of the list. The set of genes with a particularly prominent peak on the far left or right is usually the set of genes of interest.”

I would suggest move this description to the caption of Figure 3.

Experimental design

Overall, the author clearly described their experimental designs.

Validity of the findings

Pleas see below.

Additional comments

Here are some major and minor comments about the methods and findings.

Major comments:
1. Line 162-166: “A series of filters were performed …” Did the author perform any specific filtering other than default ones? If so, it should be clearly indicated.

2. Line 167-169: does the “pvalue<0.05” refer to nominal p value or adjusted p value? This question also replies to all the p values showed in the manuscript.

3. Line 182: “Differential analysis was performed using DESeq2, and differential miRNA screening conditions were: | Foldchange | ≥2 and P-value≤0.01.” The differential lncRNAs and mRNAs were identified with FC at 1.5. Is there any specific reason to use different cutoffs were used for lncRNAs, mRNAs and miRNA? Again, the author need to clarified if the p value is nominal or adjusted p value?

4. For the GO and pathway enrichment analysis, did the author use all DE genes (up- and down-regulated) as a whole input or did the enrichment analysis for up-genes and down-genes separately?

5. Line 205: the author used fdr at 0.25 as cutoff in GSEA analysis. Normally the fdr at 0.05 is commonly used. I wonder how would the results change with the fdr at 0.05.

6. Line 252-254: “Based on the selected differential genes, the hypergeometric distribution relationship between these differential genes and GO or Pathway was calculated, and a P-value was returned.” The statements might need to be rephrased. The “hypergeometric distribution relationship” is not correct. It assumes that the number of genes associated with a functional category that overlap with the set of DE genes follows a hypergeometric distribution.

7. Line 263: for pathway enrichment analysis, did the author use all DE genes or only up-regulated genes?

Minor comments:
1. Line 188: “Briefly, all genes of this species were…” Does the author mean all coding the noncoding genes in hg38? It should be clearly stated.
2. Line 237-240: since the differential miRNA had different cutoffs than lncRNA and mRNA, then the cutoffs stated in line 237 were not correct.

Reviewer 2 ·

Basic reporting

no comment

Experimental design

no commen

Validity of the findings

no comment

Additional comments

In this manuscript, the authors constructed an THP-1 macrophage infected with Talaromyces 4 marneffei model, then high-throughput sequencing of differentially expressed mRNA, LncRNA and miRNA were performed, , finally, constructing the ceRNA network of differentially expressed genes and some of the genes were identified. Although the study was designed clearly and the findings are interesting, there are several limitation of the study, which the author should address before publication;
1.First of all, in the line 58-59, the article stated that some differentially expressed genes were verified, mRNA and lncRNA both listed gene names in detail, and miRNA should also be written in detail.
2.In the line 213, the qRT-PCR primers for the validated genes must be listed; and in the abstract section, that 3 mRNAs, 5 lncRNAs and 2 miRNAs were verified in total (line 57-59), but 4 lncRNAs and 4 mRNAs were verified in the text of the article (line 307-308), which is contradictory, so, it is necessary to determine which genes have been validated.
3.In the line 208, it is not enough to use only one database for miRNA target gene prediction, several databases such as TargetScan、miRBase、miRTarBase should also be used to improve accuracy.

Reviewer 3 ·

Basic reporting

no comment

Experimental design

no comment

Validity of the findings

no comment

Additional comments

This manuscript “Identification and analysis of dysregulated lncRNA,microRNA and mRNA expression profiles and construction of ceRNA network in THP-1 macrophage infected with Talaromyces marneffei” described the sequencing of lncRNA, miRNA and mRNA in THP-1 macrophage infected with Talaromyces marneffei. These genes may be involved in immune response and inflammatory responserelated pathways and functions in TM-infected macrophages. Overall, this study is easy to follow and detail oriented, I have a few comments to improve it for further process.

1. Authors need to revise the use of terminologies and full terms and abbreviations. These abbreviations should be consistent and don't shift between the full terms and their abbreviations.
2. Despite the well design of current study, the writing of manuscript still needs further improvement.
3. In the RESULT part, some details need to be added, for example, in the ceRNA network part, the details of lncRNA-miRNA-mRNA need to be clarified.
4. In ‘qRT-PCR validation’ part, why there is not show the result of miRNA?
5. Some sentences are not needed in the DISCUSSION part, for example, line 317-325.
6. All figures are not clear enough. Please check and upload them again.

---

## Round 0.2 · accepted · Accept

Thank you for submitting to PeerJ. Please make sure, one more time to review your manuscript for any typos that might still be present in the text and to make public any dataset used in this work.

Reviewer 1 ·

Basic reporting

My questions have been addressed in the revised manuscript.

Experimental design

N/A

Validity of the findings

N/A

Additional comments

N/A

Reviewer 3 ·

Basic reporting

no comment

Experimental design

no comment

Validity of the findings

no comment

Additional comments

no comment